# Estimating Fluctuations in Neural Representations of Uncertain Environments

**Sahand Farhoodi**
Department of Mathematics and Statistics
Boston University, Boston, MA, USA
sahand@bu.edu

**Mark H. Plitt**
Department of Neurobiology
Stanford University, Stanford, CA, USA
mplitt@stanford.edu

**Lisa Giocomo**
Department of Neurobiology
Stanford University, Stanford, CA, USA
giocomo@stanford.edu

**Uri T. Eden**
Department of Mathematics and Statistics
Boston University, Boston, MA, USA
tzvi@bu.edu

## Abstract

Neural Coding analyses often reflect an assumption that neural populations respond uniquely and consistently to particular stimuli. For example, analyses of spatial remapping in hippocampal populations often assume that each environment has one unique representation and that remapping occurs over long time scales as an animal traverses between distinct environments. However, as neuroscience experiments begin to explore more naturalistic tasks and stimuli, and reflect more ambiguity in neural representations, methods for analyzing population neural codes must adapt to reflect these features. In this paper, we develop a new state-space modeling framework to address two important issues related to remapping. First, neurons may exhibit significant trial-to-trial or moment-to-moment variability in the firing patterns used to represent a particular environment or stimulus. Second, in ambiguous environments and tasks that involve cognitive uncertainty, neural populations may rapidly fluctuate between multiple representations. The state-space model addresses these two issues by integrating an observation model, which allows for multiple representations of the same stimulus or environment, with a state model, which characterizes the moment-by-moment probability of a shift in the neural representation. These models allow us to compute instantaneous estimates of the stimulus or environment currently represented by the population. We demonstrate the application of this approach to the analysis of population activity in the CA1 region of hippocampus of a mouse moving through ambiguous virtual environments. Our analyses demonstrate that many hippocampal cells express significant trial-to-trial variability in their representations and that the population representation can fluctuate rapidly between environments within a single trial when spatial cues are most ambiguous.

## 1 Introduction

The hippocampus is believed to play a significant role in spatial learning and navigation through place cells and their associated place fields [51] by constructing a spatial map of the environment [50, 52]. Furthermore, hippocampal cells are capable of representing non-spatial information such as the experience associated with places at the current time [22, 68, 70], in past [28, 46, 67], in future [1, 23, 57], and in imagination [24, 25]. Previous studies suggest that remapping, the formation of different spatial maps for different environmental contexts [26, 45, 63], can play a key role in episodic memory [11, 12, 35, 38, 40, 41, 62, 66]. However, how the remapping forms as environmental

contexts change is not completely understood yet. It has been shown that the extent of remapping depends on the magnitude of the difference between environments [19], animal's prior experience [54, 55, 66], and motivational and behavioral states such as fear conditions [17, 44, 69]. Remapping can either affect the average rates over the experiment and not the place field locations (rate remapping) [5, 12, 19, 47], or it can affect both the place field locations and the average rates over the experiment (global remapping) [12, 19, 20, 66]. It has been proposed that rate and global remapping can represent distinct hippocampal encoding systems [6, 11, 12, 39, 42], where in rate remapping the population code for space is preserved and subtle non-spatial aspects of the experience are represented through change in the expected number of spikes, while global remapping works in an all-or-none fashion and is more likely to be observed when the environmental change is substantial. Previous studies show that the observance of global or rate remapping in response to gradually accumulating changes in the environment highly depends on the training process of the animal [38, 39, 41, 56, 66]. In a recent study, Plitt & Giocomo show that with prior exposure, context-specific spatial codes tend to lead to optimal information about location in ambiguous environments [54].

There are two issues that are not fully addressed in the existing literature. First, most approaches that look at context-dependent coding assume that each neuron codes consistently for each context with a particular firing pattern. However, previous work has shown extensive trial-to-trial variability within a single context or environment where subsets of neurons may intermittently code for context on some trials, but not on others [8, 9, 30, 32, 34, 37, 55, 58, 59, 65], or form multiple maps for individual environments based on distinct contexts or reference frames (partial remapping) [12, 53, 60, 73]. Second, based on the work of Gothar et. al., different subsets of cells associated with different reference frames are not always activated simultaneously which suggests that partial remapping may result in fluctuations in the neural representation of uncertain environments, both at the population level and at the single neuron level [21]. Moreover, moment-to-moment variability within a single trial has been observed where subsets of neurons may code at certain points in time in a particular trial but not at the other points in time. The times at which a particular subset of neurons is informative may also vary from trial to trial. [27, 29, 30, 31, 32, 49].

In this paper, we present a novel state-space modeling framework [10, 33, 61] that addresses both of these issues. First, it incorporates an observation model that allows for characterizing multiple population firing patterns that code for the same environmental context. Second, it uses a state model which characterizes moment-by-moment fluctuations in the way in which the population codes for uncertain environments. This model framework provides us with a number of useful tools to address issues such as inference and estimation. In particular, it provides a decoding algorithm to estimate which of multiple environments is being represented moment-by-moment. Furthermore, this framework provides methods for performing statistical inference between different maps to determine whether or not the environment being represented by a cell population has actually changed and what is the contribution of individual cells in this process. Moreover, our approach can be used to investigate some of the previous findings, such as the effect of prior belief of an animal on the remapping pattern, at a moment-by-moment resolution.

## 2 Methods

Statistical state-space models have been successful in neural decoding in different areas, including decoding movement trajectories from the spiking activity of hippocampal place cell populations in rodents [16, 61], decoding motor cortical activity [2, 36, 64], and speech recognition and auditory attention [2, 3, 4]. Recently approaches based on state-space models have been used to estimate dynamic cognitive signals [7, 15, 71, 72]. In this section, we develop a novel state-space approach to study spatial remapping with the goal of addressing two important issues that are present in previous analyses. First, neurons can show significant trial-to-trial or moment-to-moment variability in the firing pattern used to represent the same environment. Second, in the presence of ambiguous spatial cues the environment being represented by the neural populations may fluctuate rapidly. Our state-space modeling framework addresses these two issues by comprising an observation model, which allows for coding the same environment through multiple parallel spatial maps, with a state model, that characterizes the moment-by-moment fluctuations in the population representation of uncertain environments.

In section 2.1, we explain our state-space framework in a general way which can be adapted to many studies that aim to analyze the represented environment moment-by-moment during uncertain tasks.

In section 2.2, we specify our methodology for analyzing the population activity in the CA1 region of hippocampus of mice that have developed consistent spatial maps for two original environments through random foraging, and then have been exposed to intermediate environments generated by morphing between the original environments to varying degrees. We refer to the degree of blending of these environments as the morph level.

## 2.1 State-Space framework for dynamic remapping

Let $X_1, ..., X_T$ be a discrete-time signal that is encoded by a neural population, and let $Y_1, ..., Y_T$ be the population neural response to that signal. For example, $X_t$ might represent the movement of a mouse in an environment and $Y_t$ might represent the activity of a population of place cells to that movement. We define a neural mapping between the signal and response using a conditional probability model $p(Y_1, ..., Y_T | X_1, ..., X_T)$. Equivalently, we can define the mapping as the instantaneous probability distribution of the response at each time $t$, given the full signal and the history of past responses, $p(Y_t | X_1, ..., X_T, Y_1, ..., Y_{t-1})$. Since we will always be conditioning on the signal $X_1, ..., X_T$, we will hereafter suppress writing this expression in all of our models. We will also use the notation $Y_{t_1:t_2}$ in place of $Y_{t_1}, ..., Y_{t_2}$, and so forth.

We assume that this neural mapping can change as a function of specific variables that are relevant to a particular neuroscience experiment, which we call $Z_1, ..., Z_T$. In our example of a place cell population, $Z_t$ might represent the instantaneous cognitive percept of which of multiple environments the mouse is currently occupying. Neural mappings may also vary as a function of other variables that are ancillary or unmeasured in a particular experiment, or that reflect natural variability in the population representation. Therefore, we define a set of mappings $p_k(Y_t | Z_t = j, Y_{1:t-1})$, for $k = 1, ..., K_j$, that for the particular value $Z_t = j$ describes a set of $K_j$ representations that may be reflected in the neural activity. We define a set of indicator functions $I_{j,k}(t)$ that given the value of $Z_t = j$ is equal to 1 only for the $k$ that is associated with the true mapping generating the data at time $t$. Therefore, the neural mapping that is represented at time $t$ if $Z_t = j$ is given by the observation model

$$p(Y_t | Z_t = j, Y_{1:t-1}) = \sum_{k=1}^{K_j} p_k(Y_t | Z_t = j, Y_{1:t-1}) I_{j,k}(t). \tag{1}$$

For an ensemble of neurons, the observation model becomes the product of the observation distribution functions given by Eq. 1 for individual neurons. In addition to this observation model, we construct a state transition model, $p(Z_{t+1} | Z_{1:t})$, which defines the transitions between experimentally relevant neural mappings. In the example of a place cell population, $p(Z_{t+1} | Z_{1:t})$ may reflect the probability of the mouse switching its cognitive percept of its environment due to ambiguous information.

Figure 1A depicts the state-space framework described in this section. Our goal is to construct algorithms to estimate $Z_t$ at each instant as a function of all of the past data $Y_{1:t}$ (called the filter algorithm) or as a function of all of the data over an experiment or trial $Y_{1:T}$ (called the smoother algorithm). In order to develop these algorithms, we make the following assumptions:

**(i)** $p(Y_t | Z_{1:t}, Y_{1:t-1}) = p(Y_t | Z_t, Y_{1:t-1})$

**(ii)** $p(Z_{t:T} | Z_{1:t-1}, Y_{1:t-1}) = p(Z_{t:T} | Z_{1:t-1})$

**(iii)** $p(Z_{1:t} | Z_{t+1:T}, Y_{t+1:T}) = p(Z_{1:t} | Z_{t+1:T})$.

Assumption (i) states that the distribution of the current response, $Y_t$, depends on past states only through the current state and the past responses. Assumption (ii) states that the distribution of current and future states, $Z_{t:T}$, is independent of past observations, $Y_{1:t-1}$, as long as past states, $Z_{1:t-1}$, are known. Assumption (iii) similarly states that current and past states are independent of future observations, when conditioned on future states.

As shown in Appendix 1, the filter algorithm can be constructed by moving forward in time and computing $p(Z_{1:t} | Y_{1:t})$ recursively:

$$p(Z_{1:t} | Y_{1:t}) \propto p(Y_t | Z_t, Y_{1:t-1}) p(Z_t | Z_{1:t-1}) p(Z_{1:t-1} | Y_{1:t-1}) \tag{2}$$

where $p(Y_t | Z_t, Y_{1:t-1})$ and $p(Z_t | Z_{1:t-1})$ can be computed directly from the observation model and the state model respectively, and $p(Z_{1:t-1} | Y_{1:t-1})$ is the probability distribution computed at

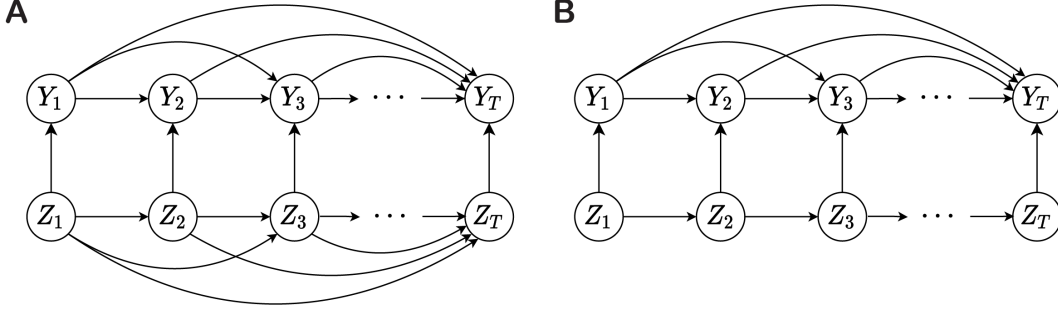

Figure 1: **A.** The relation between the state transition model and the observation model in a general case. **B.** The specific example when the state transition model is a first-order Markov chain.

the previous step of the filter algorithm. The filter distribution, $p(Z_t|Y_{1:t})$, can be computed by marginalizing over $Z_{1:t-1}$ in Eq. 2:

$$p(Z_t|Y_{1:t}) \propto P(Y_t|Z_t, Y_{1:t-1}) \sum_{Z_{1:t-1}} p(Z_t|Z_{1:t-1})p(Z_{1:t-1}|Y_{1:t-1}) \tag{3}$$

In order to estimate $Z_t$ as a function of all of the data over an experiment or trial, $Y_{1:T}$, we start from $T$ and iterate backward in time and compute $p(Z_{t:T}|Y_{1:T})$ based on its value at the next forward time step. As shown in Appendix 2, this can be expressed mathematically by

$$p(Z_{t:T}|Y_{1:T}) = \frac{p(Z_{t+1:T}|Y_{1:T})}{p(Z_{t+1:T}|Y_{1:t})} \sum_{Z_{1:t-1}} p(Z_{t+1:T}|Z_{1:t})p(Z_{1:t}|Y_{1:t}) \tag{4}$$

where $p(Z_{t+1:T}|Y_{1:T})$ is the distribution computed by the smoother algorithm at the previous step, $p(Z_{t+1:T}|Z_{1:t})$ and $p(Z_{1:t}|Y_{1:t})$ are given by the state model and the filter algorithm respectively, and as explained in detail in Appendix 2, $p(Z_{t+1:T}|Y_{1:t})$ can be computed by

$$p(Z_{t+1:T}|Y_{1:t}) = \sum_{Z_{1:t}} p(Z_{t+1:T}|Z_{1:t})p(Z_{1:t}|Y_{1:t}). \tag{5}$$

The smoothing distribution function $p(Z_t|Y_{1:T})$ can be computed by using Eq. 4 and marginalizing over $Z_{t+1:T}$. The filter and smoother algorithms explained above are here expressed generally to allow for a wide range of state and observation models. In practice, we often use simple state or observation models that can reduce the complexity of these algorithms significantly. An example of a simplification is given in section 2.2, where we use a first-order Markov chain as our state model.

### 2.2 Estimating an ambiguous environment from imaging data in a hippocampal population

In this section, we specify the framework introduced in section 2.1 to analyze spatial remapping in the CA1 region of hippocampus of mice that have developed consistent spatial maps for two original environments through random foraging, and then have been exposed to ambiguous environments that are morphed combinations of the original environments. Our data contains two-photon imaging recordings of CA1 pyramidal cells, which are collected under two different training conditions. In the "Rare" condition mice are exposed to ambiguous environments rarely, while in the "Frequent" condition they were exposed to such environments as frequent as they were exposed to the original environments. The experiment includes five different morph levels {0, .25, .5, .75, 1} where morph levels 0 and 1 are associated with the original environments, which we call "environment 0" and "environment 1" respectively throughout rest of this paper. All of the data from the Rare condition are obtained in one recording session and all of the data from the Frequent condition are obtained from a separate single recording session.

Going back to our notation in section 2.1, let $X_t$, $Y_t$, and $Z_t$ respectively be the animal's position in the virtual environment, the deconvolved activity rate estimated from the fluorescence level of each neuron [18], and the represented environment at time $t$. We assume that ambiguous environments will be represented as one of the original environments at each instant of time, and therefore $Z_t$ can be either 0 (for environment 0) or 1 (for environment 1). As demonstrated in section 3 we observed extensive trial-to-trial variability in the firing patterns within the original environments, as well as

rapid fluctuations between multiple representations in ambiguous environments. This justifies using the state-space modeling framework developed in section 2.1. As depicted in Figure 2A, for almost all neurons we observed at most two different firing patterns within the same original environment. To verify this observation, we perform hypothesis tests (GLM maximum likelihood ratio tests [13]) to determine whether more than two representations of the original environments exist. For both original environments, at the significance level of 0.01 (no multiplicity correction), the null hypothesis that only two representations exist was rejected for less than $10\%$ of the cells (see Appendix 3). Therefore, we set $K_j = 2$ in Eq. 1 for $j = 0, 1$. We observe that for each neuron, these two firing patterns either differ in the location of the place field (e.g. cell 56, morph = 0), or differ in the average firing rate (e.g. cell 13, morph = 0). In both cases we use a K-means clustering algorithm to estimate $I_{j,k}(t)$, by dividing trials into two groups based on the location of the place fields or the average firing rate. In line with recent findings [59], we assume that for each of $j = 0, 1$, $I_{j,k}(t)$ is constant across the entire trial, and is equal to 1 for the value of $k$ whose cluster contains the fluorescence signal over that trial, as determined by the K-means algorithm. For each $p_k$ in Eq. 1, we relate $Y_t$ to both $X_{1:T}$ and the past history of the neural activity over the interval $(t - h, h)$ through a history dependent, Gamma-distributed generalized linear model with identity link:

$$Y_t | (Z_t = j, Y_{1:t-1}) \sim \text{Gamma}(\mu_t = \alpha^T g_j(X_t) + \beta^T Y_{t-h:t-1}, \Sigma_j), \quad j = 0, 1 \qquad (6)$$

where $\mu_t$ is the mean of $Y_t$, $\Sigma_j$ is the dispersion parameter, which determines the variance of $Y_t$ [43], and $g_j(X_t)$ represents a set of cardinal spline basis functions that are used to approximate the neuron's place field structure [14]. A brief analysis of goodness-of-fit for this model is presented in Appendix 6. For the state transition model, we use a first-order Markov chain. This choice of state transition model simplifies the structure of our state-space framework (see Figure 1B) and as explained in Appendices 1 and 2, results in less complicated equations for the filter and smoother algorithms:

$$p(Z_t | Y_{1:t}) \propto p(Y_t | Z_t, Y_{1:t-1}) \sum_{Z_{t-1}} p(Z_t | Z_{t-1}) p(Z_{t-1} | Y_{1:t-1}) \qquad (7)$$

$$p(Z_t | Y_{1:T}) = p(Z_t | Y_{1:t}) \sum_{Z_{t+1}} \frac{p(Z_{t+1} | Z_t)}{p(Z_{t+1} | Y_{1:t})} p(Z_{t+1} | Y_{1:T}). \qquad (8)$$

In the next section we demonstrate how the statistical state-space approach described here can be used to decode the represented environment at each instant of time in ambiguous environments, both at a single cell level and at a population level. At the population level, we investigate the moment-by-moment contribution of each neuron in the spatial representation. Moreover, we use our framework to examine the effect of the prior exposure of an animal on the remapping pattern, to express the uncertainty about the decoded representations, and to perform hypothesis tests about whether multiple representations of environments are present.

## 3 Results

Before applying the state-space modeling framework described in section 2.2 to our data we used basic visualizations to demonstrate the two issues our models are designed to address: 1) many neurons contain multiple firing patterns for the same original environment, and 2) rapid fluctuations between multiple representations occur in ambiguous environments. Figure 2A shows examples of cells with spatial remapping between the original environments where the place field appears (cell 75), disappears (cell 56), or changes location (cells 220 & 13) between environment 0 and environment 1. For individual cells and within the same environment, we observe multiple parallel spatial maps. For instance, cell 56 has two spatial maps that differ in location of the place field in environment 0, and cell 13 is only active over a subset of trials in environment 0. The place cells, together, build a population-wide representation of the environments, as depicted in Figure 2B for trial 5, by sorting cells based on location of the center of their place fields during this trial. However, due to different firing patterns for different environments, the population-wide representation differs substantially for a trial from another environment (e.g. trial 1), i.e. the diagonal structure of place field centers observed for trial 5 is lost. For trials of the same environment, the population-wide representations can be similar (e. g. trials 5 and 39) or considerably different (e. g. trials 5 and 96). This emphasizes the fact that the same environment is represented through multiple firing patterns in this data. In addition to these observations, our observation models allowed us to perform hypothesis

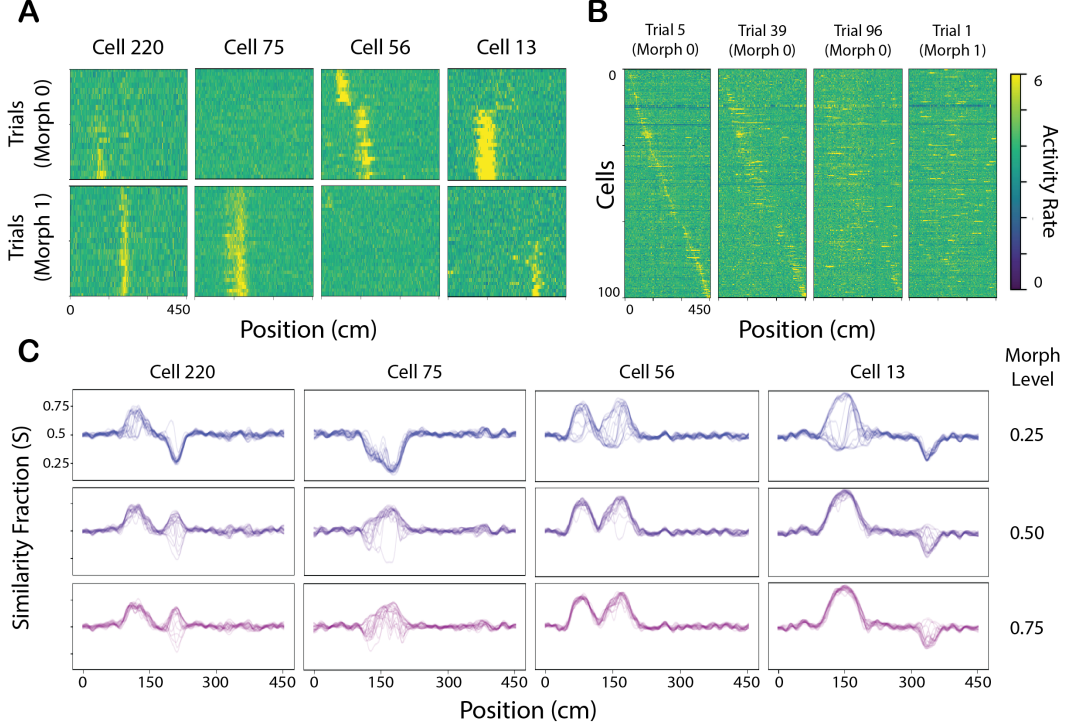

Figure 2: **A.** Each column shows the neural activity of an individual cell in the two original environments. Each row of each heat map represents the cell activity during a single trial as a function of position. Trials in each heat map are sorted with respect to the peak activity location. **B.** Each column shows the activity of 100 place cells during an individual trial as a function of position. Cells are sorted according to the location of their respective place fields for trial 5. **C.** Each row shows the similarity fraction (Eq. 9) of trials with the same morph level as a function of position. Each line in each plot corresponds to a single trial.

tests (GLM maximum likelihood ratio tests [13]) to determine whether multiple representations of the original environments are present. At the significance level of 0.01 (no multiplicity correction), the null hypothesis that only one representation exists was rejected for 94% of cells for environment 0, and for 92% of cells for environment 1 (see Appendix 3 for details). As a rough, initial measure of whether there are rapid fluctuations in the represented environment during trials with ambiguous spatial cues, we defined the similarity fraction of trial $tr$ at time $t$ by

$$S_t = \frac{\frac{1}{m_0} \sum\limits_{k \in T_0} (Y_t^{tr} - Y_t^k)^2}{\frac{1}{m_0} \sum\limits_{k \in T_0} (Y_t^{tr} - Y_t^k)^2 + \frac{1}{m_1} \sum\limits_{k \in T_1} (Y_t^{tr} - Y_t^k)^2} \tag{9}$$

where $Y_t^j$ denotes the activity rate for trial $j$ at time $t$, $T_0$ and $T_1$ denote the set of trials for environment 0 and environment 1 respectively, and $m_i = |T_i|$ for $i = 0, 1$. $S_t$ measures the relative average distance between trial $tr$ and all of the trials from the original environments at time $t$, and is used as a proxy for the environment being represented, i.e. the closer $S_t$ is to 0, more probable it is that environment 0 is represented. In Figure 2C we observe moment-to-moment fluctuations in $S$ during a single trial which suggests that there are rapid fluctuations in the neural representation of ambiguous environments. These fluctuations tend to occur mostly at place field locations. In addition, some cells exhibit dramatic changes in $S$ as a result of changing the morph level (e.g. cells 220 & 75), as well as extensive trial-to-trial variability in $S$ at certain locations for trials within the same environment (e.g. cell 13, morph = 0.25, location 100-200 cm). Furthermore, our state transition model allowed us to perform hypothesis tests (Wald tests) to determine whether the observed fluctuations are statistically significant. At the significance level of 0.01 (no multiplicity correction), the null hypothesis that there is no fluctuation in the represented environment within any trials was rejected for all cells (see Appendix 3 for details).

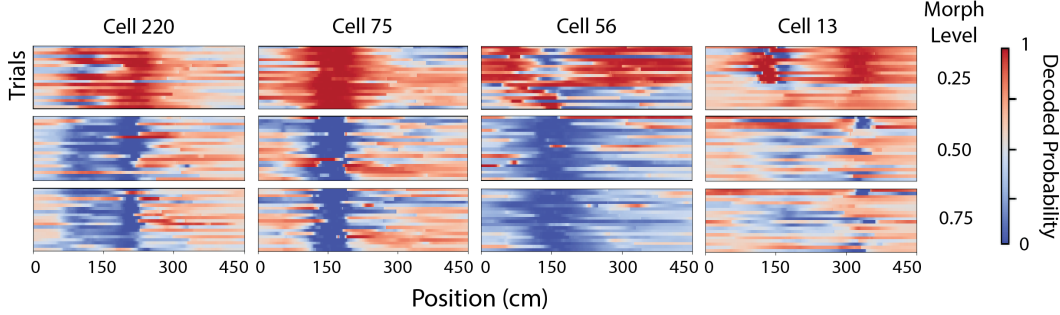

Figure 3: For each cell, the probability that environment 0 is represented, computed using the smoother algorithm, is illustrated using three different heat maps, one for each morph level. Each row of each heat map shows this probability function during a single trial as a function of position in the virtual environment.

To show the effect of incorporating multiple representations of the original environment on decoding, we fit two models, one with multiple representations ($K_j = 2$) and one with a single representation ($K_j = 1$), and compared the decoding results for trials within environments 0 and 1 (where we know the the ground truth). For each trial, the average probability (over time) of decoding the correct environment was computed. Further, for each cell, the average probability of decoding the correct environment (over all trials) was computed for the two models. Based on these analysis, we observed that the state-space model improves decoding accuracy for $87\%$ of the cells (See Appendix 4 for details).

Based on our hypothesis tests and the observations summarized in Figure 2 we have evidence for multiple population firing patterns that code for the same environment, and rapid fluctuations in the population code for ambiguous environments. Therefore, we used the state-space modeling framework defined in section 2.2 to decode the state that indicates which environment is represented, moment-by-moment. Figure 3 illustrates the output of our smoother algorithm given in Eq. 8 for the same set of cells shown in Figure 2. Based on this figure, our state-space framework allows us to capture the trial-to-trial variability in the represented environment (e.g. cell 56, morph = 0.25) as well as the rapid fluctuations in the estimated environment during individual trials (e.g. cell 13, morph = 0.25, location 100-200 cm). The decoded probabilities are more or less in accordance with what we expect to see based on the similarity fraction illustrated in Figure 2C. For instance, in both Figure 2C and Figure 3, cell 220 expresses two distinguishable peaks for each morph level located at around 100 cm and 200 cm, and cell 75 changes the represented environment from environment 0 to environment 1 at the spatial interval 120-180 cm as the morph level increases. We observe that the uncertainty in the decoded environment usually is minimized at place fields.

In Figure 4, we illustrate the results of our analysis at a population level. Figure 4A shows the decoded probability that environment 0 is represented for two small groups of cells. By comparing this figure and Figure 3 we observe that different cells code for an environment intermittently and each dominate the population code at certain locations. Therefore, for small ensembles of cells, the population code can change dramatically by replacing a cell by another (compare left and right panels for morph = 0.25, 0.5, 0.75, location 75-225 cm). In Figure 4B, the estimated environment according to the neural activity of the whole population is shown for two different training conditions, Rare and Frequent. Examples of within-trials fluctuations for individual trials are shown in Appendix 5. In accordance with previous results by Plitt & Giocomo, we see that the transition in the represented environment as the morph level increases gradually from 0 to 1 is sudden for the Rare condition and gradual for the Frequent condition (see "avg." plots) [54]. Moreover, our framework enables us to investigate this phenomenon moment-by-moment. For example, we observe that for Rare condition, the decoded probability changes gradually for some locations, e.g. 0-70 cm and 250-400 cm, and changes suddenly for other locations, e.g. 70-250 cm and 400-450 cm. Indeed, the former two intervals correspond to the beginning of the track and the reward locations respectively, and there are several reasons to expect ambiguous representations at these locations.

Next, in Figure 4C we investigate the contribution of different cells in the estimated environment by plotting the difference in the log-likelihood for the two environments based on the model in Eq. 6. We focus on two groups of cells, each of size seven, that decode the represented environment differently for all trials with morph level 0.5 (first row), and for one particular trial (second row). By looking at

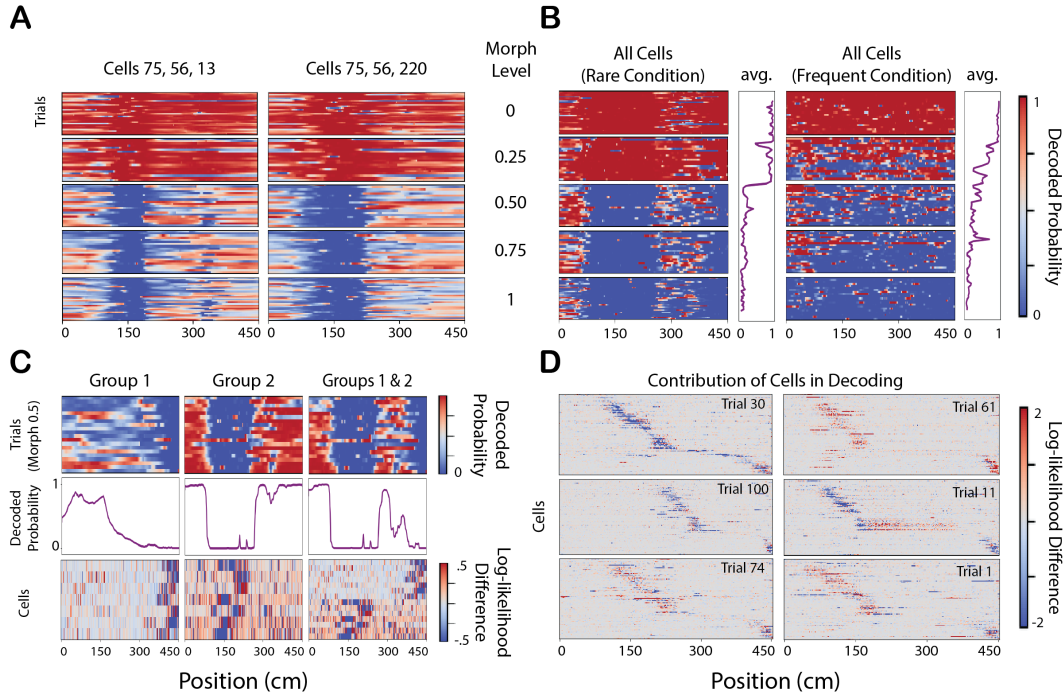

Figure 4: **A.** Each column illustrates the decoded probability for a group of three cells in five heat maps, one for each morph level. **B.** The full-population decoded probability for two different training conditions. For each condition, moment-by-moment decoded probability is shown on left and full-length decoded probability for different trials is shown on right. **C.** The first row shows the population decoded probability for three different groups of cells for trials with morph level 0.50. The second row illustrates the decoded probability only for trial 30. The third row shows the contribution of different cells within each group by showing the difference in the log-likelihood of their activity between environment 0 and environment 1. **D.** Shows the contribution of 100 cells in population-wide decoding as a function of position for 6 different trials in ambiguous environments. In all heat maps, cells are sorted according to the location of lowest log-likelihood difference for trial 30.

the difference in log-likelihood from each cell (third row) we observe that in group 1 all cells code for environment 0 weakly during 0-180 cm, but the representation slowly shifts to environment 1 by positions 180-450 cm. In group 2 however, cells represent environment 1 in the region 100-250 cm strongly and represent environment 0 outside this interval. In the last column, we combine these two groups and see that during different intervals different groups dominate the population code. For instance, between 0-100 cm both groups code for environment 0 which is followed by a shift in the represented environment (100-250 cm) as group 2 cells dominate the population code. Beyond that, the two groups seem to be at odds in their representations (250-380 cm) with the group 1 cells eventually dominating the population code, leading to a confident decode of environment 1 between 380-450 cm. In Figure 4D we look at the contribution of different cells in the population code for 6 different trials with ambiguous spatial cues. We observe that the information in the population activity about the represented environment is often dominated by a small subset of cells, and further, this subset can change moment-by-moment and trial to trial (compare trials 30 and 61 for locations 50-120 cm, or trials 30 and 100 for locations 200-300 cm). In addition, we observe that each cell can dominate the population code at different locations for different trials (note the horizontal shift of blue shade for trials 30 and 100).

To justify the advantages of the model structure proposed in this paper, we compared our approach with 2 naive approaches: (1) one using only a single spatial map (setting $K_j = 1$) for all cells, and (2) one based directly on the likelihood of each map, without the state-space model. Using approach (2), we observed that the decoded probabilities for each cell were noisier, with means hovering around .5, and large standard deviations of about 0.35. This issue, which is partly due to the low signal-to-noise ratio of the data, results in increased uncertainty about the represented environment, and would not be alleviated with simple fixes such as kernel methods. At the population level, this noisiness drops significantly, but is still substantial, with an average standard deviation around 0.22. The average

probability of decoding the correct environment during the original, unambiguous environments for approaches (1) and (2) were computed and the difference between these values and the same values for our state-space modeling approach is shown in Appendix 4. This suggests that the state-space modeling framework improves decoding accuracy in 87% of cells, and naive methods cannot yield similar results, such as those shown in Fig. 4.

## 4    Discussion

In this paper, we developed a novel state-space framework to capture two specific features of neural remapping that are not addressed in previous analyses. First, neural populations may exhibit significant trial-to-trial and moment-to-moment variability in the firing patterns used to represent a given environment or stimulus. Second, in ambiguous environments neural populations may rapidly fluctuate between multiple representations. The state-space framework comprises a state transition model that describes how rapidly the representation can fluctuate, and an observation model that can include multiple ways of representing the same environment. We applied our methods to the population activity in the CA1 region of hippocampus of mice that have developed consistent spatial maps for two original environments and then are exposed to intermediate ambiguous environments. Our analyses demonstrated that the two features in the neural representation which we aim to address are present in this data and our methods are capable of capturing them. The model-based nature of our methods provides us with a set of statistical tools for performing hypothesis tests and investigating the contribution of individual cells in the population code.

To the best of our knowledge, previous approaches for understanding neural coding of ambiguous environments have not explicitly included multiple spatial maps and models of transitions between those maps. Our approach makes specific assumptions about the dynamics of hippocampal representations in order to increase the statistical power of our inferences. Our data analysis results demonstrated the specific advantages of using a model structure that incorporates these assumptions, e. g. improvement in decoding in known environments for 87% of cells, and substantial reduction in the uncertainty of the decoded environment.

While the methods in section 2.1 are developed for general classes of models, our implementation of these methods in section 2.2 made a number of simplifying assumptions, that may not hold in general. We used a simplified model that assumed that every cell had exactly two representations of each environment. We also assumed that the different original environments do not share the same representation. Future work will focus on more complicated state-space model structures to account for any number of representations across multiple environments. Furthermore, we assumed that while the representation of the environment could switch within a trial, which of the two representations within an environment was expressed remained the same across each trial. We used a simple K-means algorithm to determine which of these representations was present for each trial, which was determined independently of the estimate of the environment being represented. If these assumptions do not hold, the broader framework described in 2.1 can still be used to estimate the dynamical structure of shifting neural representations. If the number of representations of a single environment is unknown and differs across environments, the state-space model structure can be used as a component of an Expectation-Maximization (EM) algorithm [48] to estimate the number of representations in each environment, and to determine which of those representations is present at each moment or trial.

## Broader Impact

Most classical neural coding analyses heavily depend on the simplifying assumptions that neurons respond uniquely and consistently to particular stimuli. Such assumptions may be valid for simple stimuli that are clearly distinguishable, leading to population codes where each neuron provides consistent information about each stimulus across multiple presentations. In such scenarios, we might expect downstream brain regions to decode the represented stimuli in such tasks by averaging the information over a large set of upstream neurons. However, as the class of neuroscience experiments becomes more complicated, and we encounter a higher level of uncertainty in the population code, where subsets of neurons code intermittently for a stimulus based on other uncontrolled factors, new methods that account for moment-by-moment temporal dynamics and the multiple ways in which neural populations can represent information across different stimulus become increasingly important. The resulting analyses may then provide added insight into the ways in which the brain resolves

ambiguous stimuli, or represents specific signals in the face of changing situations and distractions. As neuroscience experiments begin to explore more naturalistic tasks and stimuli, and reflect more ambiguity in neural representations, neural data analysis methods, such as the one presented in this paper, must adapt to reflect these features.

## Acknowledgments and Disclosure of Funding

The work for this project was supported by the Simons Collaboration on the Global Brain 542971 and the National Institute of Mental Health MH105174 to UTE, funding from the New York Stem Cell Foundation, Office of Naval Research N00141812690, Simons Foundation 542987SPI, the Vallee Foundation and the James S McDonnell Foundation to LMG, and a National Science Foundation Graduate Research Fellowship awarded to MHP.

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
