[Supplementary Material]

# Supplementary Material: Estimating Fluctuations in Neural Representations of Uncertain Environments

**Sahand Farhoodi**
Department of Mathematics and Statistics
Boston University, Boston, MA, USA
sahand@bu.edu

**Mark H. Plitt**
Department of Neurobiology
Stanford University, Stanford, CA, USA
mplitt@stanford.edu

**Lisa Giocomo**
Department of Neurobiology
Stanford University, Stanford, CA, USA
giocomo@stanford.edu

**Uri T. Eden**
Department of Mathematics and Statistics
Boston University, Boston, MA, USA
tzvi@bu.edu

## Appendix 1: Derivation of the filter algorithm

In order to develop filter and smoother algorithms we make the following assumptions

$$\textbf{(i) } p(Y_t|Z_{1:t}, Y_{1:t-1}) = p(Y_t|Z_t, Y_{1:t-1}) \tag{S1}$$

$$\textbf{(ii) } p(Z_{t:T}|Z_{1:t-1}, Y_{1:t-1}) = p(Z_{t:T}|Z_{1:t-1}) \tag{S2}$$

$$\textbf{(iii) } p(Z_{1:t}|Z_{t+1:T}, Y_{t+1:T}) = p(Z_{1:t}|Z_{t+1:T}) \tag{S3}$$

Equations S2 and S3 imply that

$$p(Z_t|Z_{1:t-1}, Y_{1:t-1}) = p(Z_t|Z_{1:t-1}) \tag{S4}$$

$$p(Z_t|Z_{t+1:T}, Y_{t+1:T}) = p(Z_t|Z_{t+1:T}). \tag{S5}$$

Additionally, if the state transition model is a first-order Markov chain we have:

$$p(Z_{t:T}|Z_{1:t-1}) = p(Z_{t:T}|Z_{t-1}) \tag{S6}$$

$$p(Z_t|Z_{1:t-1}) = p(Z_t|Z_{t-1}) \tag{S7}$$

The filter algorithm can be derived as follows:

$$
\begin{aligned}
p(Z_{1:t}|Y_{1:t}) &\propto p(Y_t|Z_{1:t}, Y_{1:t-1})p(Z_{1:t}|Y_{1:t-1}) && \text{Bayes rule} \\
&\propto p(Y_t|Z_t, Y_{1:t-1})p(Z_{1:t}|Y_{1:t-1}) && \text{From Eq. S1} \\
&\propto p(Y_t|Z_t, Y_{1:t-1})p(Z_t|Z_{1:t-1}, Y_{1:t-1})p(Z_{1:t-1}|Y_{1:t-1}) && \text{Product rule} \\
&\propto p(Y_t|Z_t, Y_{1:t-1})p(Z_t|Z_{1:t-1})p(Z_{1:t-1}|Y_{1:t-1}) && \text{From Eq. S4}
\end{aligned}
$$

Next, the probability function $p(Z_t|Y_{1:t})$ can be computed by marginalizing over $Z_{1:t-1}$:

$$p(Z_t|Y_{1:t}) \propto p(Y_t|Z_t, Y_{1:t-1}) \sum_{Z_{1:t-1}} p(Z_t|Z_{1:t-1})p(Z_{1:t-1}|Y_{1:t-1})$$

**Special case: first-order Markov chain**

If the state transition model follows a first-order Markov chain, then the filter algorithm can be simplified as follows:

$$p(Z_t|Y_{1:t}) \propto p(Y_t|Z_t, Y_{1:t-1}) \sum_{Z_{1:t-1}} p(Z_t|Z_{1:t-1})p(Z_{1:t-1}|Y_{1:t-1})$$

$$\propto p(Y_t|Z_t, Y_{1:t-1}) \sum_{Z_{1:t-1}} p(Z_t|Z_{t-1})p(Z_{1:t-1}|Y_{1:t-1}) \qquad \text{From Eq. S7}$$

$$\propto p(Y_t|Z_t, Y_{1:t-1}) \sum_{Z_{t-1}} p(Z_t|Z_{t-1}) \sum_{Z_{1:t-2}} p(Z_{1:t-1}|Y_{1:t-1})$$

$$\propto p(Y_t|Z_t, Y_{1:t-1}) \sum_{Z_{t-1}} p(Z_t|Z_{t-1})p(Z_{t-1}|Y_{1:t-1}) \qquad \text{Law of total probability}$$

## Appendix 2: Derivation of the smoother algorithm

The smoother algorithm can be derived as follows:

$$
\begin{aligned}
p(Z_{t:T}|Y_{1:T}) &= p(Z_{t+1:T}|Y_{1:T})p(Z_t|Z_{t+1:T}, Y_{1:T}) & \text{Product rule} \\
&= p(Z_{t+1:T}|Y_{1:T})p(Z_t|Z_{t+1:T}, Y_{1:t}) & \text{From Eq. S5} \\
&= p(Z_{t+1:T}|Y_{1:T}) \sum_{Z_{1:t-1}} p(Z_{1:t}|Z_{t+1:T}, Y_{1:t}) & \text{Law of total probability} \\
&= p(Z_{t+1:T}|Y_{1:T}) \sum_{Z_{1:t-1}} \frac{p(Z_{t+1:T}|Z_{1:t}, Y_{1:t})p(Z_{1:t}|Y_{1:t})}{p(Z_{t+1:T}|Y_{1:t})} & \text{Bayes rule} \\
&= \frac{p(Z_{t+1:T}|Y_{1:T})}{p(Z_{t+1:T}|Y_{1:t})} \sum_{Z_{1:t-1}} p(Z_{t+1:T}|Z_{1:t})p(Z_{1:t}|Y_{1:t}) & \text{From Eq. S2}
\end{aligned}
$$

The denominator can be computed by conditioning on $Z_{1:t}$:

$$
\begin{aligned}
p(Z_{t+1:T}|Y_{1:t}) &= \sum_{Z_{1:t}} p(Z_{t+1:T}, Z_{1:t}|Y_{1:t}) & \text{Law of total probability} \\
&= \sum_{Z_{1:t}} p(Z_{t+1:T}|Z_{1:t}, Y_{1:t})p(Z_{1:t}|Y_{1:t}) & \text{Product rule} \\
&= \sum_{Z_{1:t}} p(Z_{t+1:T}|Z_{1:t})p(Z_{1:t}|Y_{1:t}) & \text{From Eq. S2}
\end{aligned}
$$

The probability function $p(Z_t|Y_{1:T})$ can be computed by marginalizing over $Z_{t+1:T}$.

**Special case: first-order Markov chain**

If the state transition model follows a first-order Markov chain, then the smoother algorithm can be simplified as follows:

$$
\begin{aligned}
p(Z_{t:T}|Y_{1:T}) &= \frac{p(Z_{t+1:T}|Y_{1:T})}{p(Z_{t+1:T}|Y_{1:t})} \sum_{Z_{1:t-1}} p(Z_{t+1:T}|Z_{1:t})p(Z_{1:t}|Y_{1:t}) \\
&= \frac{p(Z_{t+1:T}|Y_{1:T})}{p(Z_{t+1:T}|Y_{1:t})} \sum_{Z_{1:t-1}} p(Z_{t+1:T}|Z_t)p(Z_{1:t}|Y_{1:t}) & \text{From Eq. S6} \\
&= \frac{p(Z_{t+1:T}|Y_{1:T})}{p(Z_{t+1:T}|Y_{1:t})} p(Z_{t+1:T}|Z_t)p(Z_t|Y_{1:t}) & \text{Law of total probability}
\end{aligned}
$$

Therefore, the marginal distribution $p(Z_t|Y_{1:T})$ can be computed as

$$
\begin{aligned}
p(Z_t|Y_{1:T}) &= \sum_{Z_{t+1:T}} p(Z_{t:T}|Y_{1:T}) && \text{Law of total probability}\\[6pt]
&= \sum_{Z_{t+1:T}} \frac{p(Z_{t+1:T}|Y_{1:T})}{p(Z_{t+1:T}|Y_{1:t})} p(Z_{t+1:T}|Z_t) p(Z_t|Y_{1:t})\\[6pt]
&= p(Z_t|Y_{1:t}) \sum_{Z_{t+1:T}} \frac{p(Z_{t+1:T}|Y_{1:T})}{p(Z_{t+1:T}|Y_{1:t})} p(Z_{t+2:T}|Z_{t+1}) p(Z_{t+1}|Z_t) && \text{Product rule and Eq. S7}\\[6pt]
&= p(Z_t|Y_{1:t}) \sum_{Z_{t+1}} p(Z_{t+1}|Z_t) \sum_{Z_{t+2:T}} \frac{p(Z_{t+1:T}|Y_{1:T})p(Z_{t+2:T}|Z_{t+1})}{p(Z_{t+1:T}|Y_{1:t})}\\[6pt]
&= p(Z_t|Y_{1:t}) \sum_{Z_{t+1}} p(Z_{t+1}|Z_t) \sum_{Z_{t+2:T}} \frac{p(Z_{t+1:T}|Y_{1:T})p(Z_{t+2:T}|Z_{t+1})}{p(Z_{t+2:T}|Z_{t+1},Y_{1:t})p(Z_{t+1}|Y_{1:t})} && \text{Product rule}\\[6pt]
&= p(Z_t|Y_{1:t}) \sum_{Z_{t+1}} \frac{p(Z_{t+1}|Z_t)}{p(Z_{t+1}|Y_{1:t})} \sum_{Z_{t+2:T}} \frac{p(Z_{t+1:T}|Y_{1:T})p(Z_{t+2:T}|Z_{t+1})}{p(Z_{t+2:T}|Z_{t+1})} && \text{From Eq. S2 \& S6}\\[6pt]
&= p(Z_t|Y_{1:t}) \sum_{Z_{t+1}} \frac{p(Z_{t+1}|Z_t)}{p(Z_{t+1}|Y_{1:t})} p(Z_{t+1}|Y_{1:T}) && \text{Law of total probability}
\end{aligned}
$$

## Appendix 3: Hypothesis Tests

### Hypothesis test for multiple representations of the original environments

The observation model described in equation 1 is:

$$
p(Y_t|Z_t = j, Y_{1:t-1}) = \sum_{k=1}^{K_j} p_k(Y_t|Z_t = j, Y_{1:t-1}) I_{j,k}(t). \tag{S8}
$$

To test if multiple representations of the original environments exist, for each $j$ we define the null and alternative hypotheses as follows

$$
H_0 : K_j = 1 \qquad H_1 : K_j = 2
$$

and use the GLM maximum likelihood ratio test procedure [1]. The likelihood ratio test statistic for the null hypothesis is given by

$$
\lambda_{LR} = -2\ln\left[\frac{\sup_{K_j=1} p(Y_{1:T}|X_{1:T})}{\sup_{K_j=2} p(Y_{1:T}|X_{1:T})}\right]. \tag{S9}
$$

where the terms on the numerator and denominator are maximum likelihoods computed under hypotheses $H_0$ and $H_1$ respectively. The p-value for this test is computed by comparing the observed statistic, $\lambda_{LR}$, to a $\chi^2$ distribution with degrees-of-freedom equal to the difference between the number of parameters in the models used in the numerator and denominator of equation S9. With the significance level $\alpha = 0.01$ (with no multiplicity correction) the null hypothesis was rejected for 94% of cells when $j = 0$, and for 92% of cells when $j = 1$.

To test if more than two representations of the original environments exist, for each $j$ we define the null and alternative hypotheses as

$$
H_0 : K_j = 2 \qquad H_1 : K_j > 2 \tag{S10}
$$

and use the GLM maximum likelihood ratio test procedure [1] as explained above. With the significance level $\alpha = 0.01$ (with no multiplicity correction) the null hypothesis was rejected for less than 10% of the cells.

### Hypothesis test for state transition model

In the framework specified in section 2.2, we use a first-order Markov chain with two states as our state transition model. Let $q$ be the probability of going from state 0 (corresponding to environment

0) to state 1 (corresponding to environment 1). In order to investigate if the fluctuations in the represented environment in the presence of ambiguous spatial cues are statistically significant, we consider the following hypotheses:

$$H_0 : q = 0 \qquad H_1 : q > 0$$

We perform a Wald test using a normal approximation to the posterior distribution of $q$ given the data. By assuming a uniform prior distribution for $q$ the posterior distribution can be computed by

$$p(q|\text{Data}) \propto p(\text{Data}|q) = \prod_{t=1}^{T} p(Y_t|q) = \prod_{t=1}^{T} \Big[ \sum_{Z_t} p(Y_t|Z_t, q) p(Z_t|q) \Big]$$

where $p(Y_t|Z_t, q)$ is given by equations 1 and 6, and $p(Z_t|q)$ can be computed by the output of the smoother algorithm given in equation 8. The normal approximation is justified based on the posterior distribution of $q$ depicted in Fig. S1. With the significance level $\alpha = 0.01$ (with no multiplicity correction) the null hypothesis was rejected for all cells.

Figure S1: For four different cells, the posterior distribution function is computed and depicted. Based on the shape of these distribution functions, we used a Normal approximation to perform Wald tests.

## Appendix 4: Comparative analysis for two naive approaches

To explore the advantages of the model structure proposed in this paper, we compared our approach with 2 naive approaches. The analyses explained below suggest that our approach improves the decoding accuracy substantially, and these naive approaches cannot yield similar results to those shown in Fig. 4 of the paper.

**Approach 1: using only a single spatial map**

In this approach, we set $K_j = 1$ for all cells and the effect of this assumption on the decoding results. Here, we concentrate only on trials within original environments, where we know the correct environment and hence can assess how well is the decoding. For each trial, the average probability (over time) that the correct environment has been decoded is computed using the state-space modeling framework and approach 1. Figure S2A shows the results for cells 13 and 56 (two cells that clearly have multiple representations based on Fig. 2 of the paper), which clearly indicate that incorporating multiple representations improves decoding substantially. Further, for every cell in the population, the average probability of decoding the correct environment during the original, unambiguous environments for this approach is computed and the difference between these values and the same values when our full state-space modeling approach is used is shown in Fig. S2B. Based on this analysis, we observed that the state-space model improves decoding accuracy for $87\%$ of the cells.

**Approach 2: decoding directly based on the likelihood of each map, without the state-space model**

In this approach, instead of using a state-space structure, we use the likelihoods given by Eq. (1) of the paper directly to decode the represented environment. Using this approach, we observed that the decoded probabilities for each cell were noisier, with means hovering around .5, and large standard deviations of about 0.35. This issue, which is partly due to the low signal-to-noise ratio of the data, results in increased uncertainty about the represented environment, and would not be alleviated with simple fixes such as kernel methods. At the population level, this noisiness drops significantly, but is still substantial, with an average standard deviation around 0.22. For each cell, the average

probability of decoding the correct environment during the original, unambiguous environments for this approach is computed and the difference between these values and the same values when our full state-space modeling approach is used is shown in Fig. S2B. Based on this analysis, we observed that the state-space model improves decoding accuracy for $87\%$ of the cells.

Figure S2: **A.** Each plot shows a histogram of the average probability (over time) of correctly decoding the trials within unambiguous environments. The orange histogram shows this for the full state-space model and the blue one shows the same histogram for Approach 1. **B.** The average probability of decoding the correct environment during the original, unambiguous environments for approaches 1 and 2 were computed, and the difference between these values and the same values when our full state-space modeling approach is used is shown. For each plot, cells are ordered in a way that the plot is increasing (the order of the cells is not the same for blue and orange plots).

## Appendix 5: Examples of within-trial fluctuations for individual trials at the population level

Fig. S3 shows the decoded environment for a few sample trials based on the neural activity of the whole population. In some trials (e. g. trials 65 & 25) we observe few fluctuations, while in other trials (e. g. trial 2) several rapid fluctuations are observed.

Figure S3: Examples of within-trial fluctuations for three individual trials at the population level.

## Appendix 6: Goodness-of-fit for Gamma models

In Eq. 6, we use a history dependent, gamma-distributed generalized linear model with identity link. Figure S4 illustrates the quality of this fit for a couple of neurons. In addition, an analysis of deviance for all of the cells in the population did not show statistically significant excessive deviance for the fit model for any of the neurons.

Figure S4: For two sample cells, the data points (red), the Gamma fit (blue), and the 95% confidence band (orange) are depicted. These plots suggest that the Gamma model captures the visible structure in the data.

## References

[1] G. M. Cordeiro. Improved likelihood ratio statistics for generalized linear models. *Journal of the Royal Statistical Society: Series B (Methodological)*, 45(3):404–413, 1983.