[Reviews · NeurIPS 2020]

Review 1

Summary and Contributions: The authors suggest a statistical state-space model to infer latent variables that exist in addition to position information in hippocampal data. The authors examine a scenario where the neural activity fluctuates between several alternative representations while being in the same environment. They then decode the active representation at each moment in time – both for the original environments and for intermediate (morphed) environments. The analysis shows that neural representation in the same environment changes between different trials. Furthermore, the analysis suggests that representations might change within single trials.

Strengths: The results highlight the fact that multiple representations might exist for the same environment (in line with recent results by Sheintuch et al, Current Biology 2020). Incorporating multiple representations into a decoding model is, to the best of my knowledge, a novel contribution.

Weaknesses: It is not clear how allowing for multiple representations of the same environment contributes to decoding. While it is clear that the framework takes this into account, there is no analysis of the effect this has on decoding. One of the main claims – rapid fluctuations (within trial) of representations – is only weakly supported. The analysis seems very sensitive to the signal-to-noise ratio, but this is not sufficiently discussed.

Correctness: The claim of multiple representations is shown in Figure 2, but no statistics are provided about its prevalence. This phenomenon has been reported elsewhere (Sheintuch et al), so is probably correct. The claim of rapid fluctuations seems weakly supported. Figure 2 shows within-trial fluctuations, but these are based on single cells. This implies high sensitivity to SNR. When analyzing the full population, the rapid fluctuations seem to almost vanish (Figure 4B). This is also in line with the reports of Sheintuch et al, where transitions were only between trials.

Clarity: The paper is clearly written, and the methods are explained well. One point that requires more elaboration is the definition of I_j,k and K_j. For instance, in equation 1 is the same p_k used for different js? It would be useful to include a sketch of z_t and I_jk in figure 1.

Relation to Prior Work: The work of Sheintuch et al is highly relevant and not mentioned here. Also unmentioned are the papers mentioned in the introduction of this work that describe multiple representations of the same environment.

Reproducibility: Yes

Additional Feedback: POST REBUTTAL UPDATE: The authors answered some of my concerns, but not all. Specifically, they showed that incorporating multiple representations improves decoding, and that representations fluctuate within single trials. Although I'm left unsure about the interplay between fluctuations - in a single trial of a morph, do representation oscillate between single versions of the base environments or between all combinations? I am still worried that most of the results are from single cell analyses, which are greatly affected by low SNR. And in the population analysis - there is no mention of the effect of SNR, although it seems to matter quite a lot. Overall - I raise my score from 5 to 6 -------------------------------------------------- 1. L170-171: Were other values of K tested for k-means? Were all cells place cells? 2. Figure 2: it would be helpful to flip the order of rows in panel C and include the 0,1 morph levels. This would make the similarity fraction point in a manner consistent with row ordering. 3. L218: “moment-to-moment fluctuations”. The figure shows S as a function of position, and not of time. It would be helpful to also view this as a function of time. 4. L263: “shifts to environment 1 by positions 180-450” . Is each trial a single sweep? The wording of the sentence suggests events that unfold as a function of time, and not of space. 5. Figure 4B: Is there a relation between the positions with ambiguity (around 300) and the SNR? It’s hard to see, but figure 2B seems to show less place fields there.


Review 2

Summary and Contributions: The authors propose a new type of state-space modelling framework, specifically meant to address issues with variability if neural firing, and ambiguous environments, The performance of the decoder is tested on a data set from hippocampal place cells.

Strengths: - The background is well described, and the model is described in a good amount of detail - The contribution seems interesting and promising - The writing is good and the figures clear - This is certainly relevant to the NeurIPS community and seems novel

Weaknesses: - Lack of comparison with other algorithms (see below)

Correctness: - I am not an expert on this area, but I did not spot any errors

Clarity: - Clear and well written

Relation to Prior Work: - It would be useful to have comparison to other models, e.g. how this relates to other types of bayesian smoothing and filtering, hidden markov models etc. A direct comparison with the output of other models would have been even better- While the conditioning on X is stated to be implicit, it might still be preferable to have it part of the notation as a reminder, e.g. with subscript X

Reproducibility: Yes

Additional Feedback: - While the conditioning on X is stated to be implicit, it might still be preferable to have it part of the notation as a reminder, e.g. with subscript X


Review 3

Summary and Contributions: This submission presents a state-space-based modeling framework to study the hippocampal remapping. This method allows instantaneous estimation of the environment represented by the neural population. The authors applied the algorithm to analyze the calcium imaging data collected when the mouse was moving though ambiguous virtual environment under different morph levels. The authors found that the states represented in the neural population can rapidly fluctuate even within a single trial.

Strengths: I enjoyed reading the paper. The probabilistic framework to study remapping is quite appealing. Previously most remapping analyses are done by taking the averaged firing rate maps at the time scale of minutes (except for a few studies). The analysis of the hippocampal calcium imaging dataset is quite detailed. The results presented in Fig 2,3,4 are scientifically interesting. In particular, I found Fig 4B,C to be informative.

Weaknesses: * While I appreciate the probabilistic approach in the paper, the algorithm seems to be a quite straight-forward extension of previous state-space models (HMMs). So I feel the algorithm by itself does not carry too much novelty. * The results from this algorithm is not compared to alternative methods. Would simple methods, such as PCA, also yield similar results shown in Fig 4B,C,D? It would be helpful to justify the necessity of using such more complicated method to analyze these data. The applicability of the proposed method seems to be quite restrictive. The paper would be stronger if the authors could demonstrate or propose some other potentially applications. Overall, I feel that the scientific results presented upon further substantiation (based on larger sample size) could result in a solid scientific paper in a scientific journal. It’s unclear to me how appealing the current manuscript would be for the NeurIPS audience partly due to the incremental technical contributions. Another technical issue: the Gamma model in Eq (6) is likely a poor model for the deconvolve calcium response. It would be useful to show how well the model can fit the data. ***************modified after rebuttal I thank the authors for the feedback. After seeing everything, I remain slight negative about this manuscript. But I also feel this is a borderline paper- perhaps the arguments could be made either way. I am still slightly negative, because I think i) the algorithmic contribution is too incremental; ii) I am not convinced the benefit/gain of modeling the transition probability, and whether one could already obtains most of the results at the population level by running simple procedure such as PCA etc. The authors' rebuttal did not fully address these issues.

Correctness: The claims and method appears to be sound.

Clarity: The paper is well written and relatively easy to follow.

Relation to Prior Work: yes, the relation to the previous work is well described.

Reproducibility: Yes

Additional Feedback: More detailed comments: * Line 204-206. Is trial 5 and 96 from the same recording session? * It is a bit odd that Fig 1 does not contain X_i. It would be useful to add that in. * I notice that ref [30] is now published as a paper at Cell. It would be good to update the reference. * The title seems to be too vague. It might be beneficial to replace it with a more specific title.


Review 4

Summary and Contributions: This paper explores the significance of trial to trial variability in neural responses, and also moment to moment variability within a single trial, particularly in uncertain environments. The paper proposes a state-space modeling framework to address both issues, and then interprets neurophysiological data in terms of the framework.

Strengths: The neurophysiological phenomenon being addressed seems very important, and computationalists need to be more aware of the variability in neural responses. The theoretical framework presented proposes on interpretation of data that seem otherwise noisy. The theory itself seems sensible.

Weaknesses: Although the model is a nice instantiation of a qualitative theory, I'm uncertain what value the model adds beyond the qualitative theory. The primary validation of the model is the fact that its interpretations seem to be consistent with the level of uncertainty in the environment (the morph level). I'm left with lots of questions: What implications does this model have for cognition more broadly? Can the model predict phenomena it wasn't explicitly designed to explain? If brains are constantly inferring a latent representation that characterizes the environment, where does this representation live in the brain? I can't tell whether the model is a mechanism designed to explain data or whether it is supposed to give insight into computation in the brain. I'd hope a NeurIPS paper would do the latter and would be more enthusiastic about the paper if it was able to step back from the nitty gritty of the data and give a bigger picture view of why a non-neuroscientist should care about the phenomena and the model. An alternative hypothesis I wonder about is whether the brain has intrinsic stochasticity, as the model seems to indicate, or whether there are factors driving switches that are presently unaccounted for. For example, could the trial-to-trial variation be driven by recency effects (experience on the last few trials) in a systematic manner? Would such sequential dependencies be evidence against the current model?

Correctness: The work seems well carried out, although I did not look at the appendix to evaluate the details of the modeling.

Clarity: The paper is very well written. I realize that there are only 8 pages, but details of the experiment are sketchy, which made it challenging to understand exactly what the data represent. It would have helped to explain the experiment and present the data early in the paper to help motivate the model and make its purpose more concrete.

Relation to Prior Work: The framing of the work in terms of existing literature is outstanding. Very clear.

Reproducibility: Yes

Additional Feedback:

[Author Response · NeurIPS 2020]

We thank all reviewers for their careful reviews and many positive comments. Here, we address major questions and concerns that were
raised by the reviewers. We feel that the typos and minor issues are easily addressable and will be corrected.

**R1: Statistics for the prevalence of multiple representations and its effect on decoding.** In addition to Fig. 2 that illustrates the
existence of multiple representations of the same environment for a few sample neurons, we performed hypothesis tests for multiple
representations for all neurons (Appendix 3 first part) and summarized the results in L209-211 of our paper. We will add further
discussion of these results to the main body of the paper. Following the suggestion of R1, we added new analyses to show the effect of
incorporating multiple representations on decoding as follows. We fit two models, one with multiple representations ($K_j = 2$) and one
without ($K_j = 1$), and then compare the decoding results for trials within original environments (where we know the ground truth). For
each trial, the average probability (over time) that the correct environment has been decoded is computed. Fig. 5A (below) shows the
results for cells 13 and 56 (two cells that clearly have multiple representations based on Fig. 2) which clearly indicate that incorporating
multiple representations improves decoding significantly. Further, for each cell, the average probability of the correct environment (over
trials) is computed for the two models and the difference is depicted in Fig. 5C (curve indicated by "Approach (1)"). This shows that the
state space model improves decoding accuracy for 87% of cells. We will incorporate this analysis into a revision of the paper.

**R1: The existence of fluctuations between representations and its apparent conflict with the work of Sheintuch et. al. These**
**fluctuations seem to vanish for the full population.** We thank R1 for bringing this highly related work to our attention. We will
certainly add a discussion of the relation of this work to the paper. We believe that our results are not in conflict with those of this paper,
but rather compliment them. That work focuses on environments for which mice have previously developed spatial maps. In that case,
they find that a single spatial map is activated over any trial. Similarly, our model explicitly assumes that the representations over the
trained environments (morph 0 and 1) are fixed over each trial (L172-174). However, we propose that for "ambiguous" environments,
where mice have not yet developed a consistent spatial map, there may be rapid fluctuations between the original environments being
represented. Some previous studies have pointed out this phenomenon ([21, 27, 29, 30, 31, 32, 49]) and in addition, we performed
hypothesis tests for this (Appendix 3 second part). Furthermore, our analysis shows that these rapid fluctuations can happen at the
population level as well. This is hard to see in Figure 4, so we will add a panel (Fig 5b below) that illustrates these population-level
transitions over individual trials. **R1: Other values of $K$ tested? All cells place cells?** Yes, we will mention these in our revision.

**R2, R3: Lack of comparison with other algorithms and methods.** As mentioned by R1 and R3, to the best of our knowledge, there
is no existing algorithm that explicitly models multiple spatial maps, models the dynamics of transitions between those maps, and uses
real data to decode the moment-by-moment representations. To justify the advantage of this model structure, we will add a comparative
analysis to the paper, where our current approach is compared with 2 naive approaches: (1) one using only a single spatial map (setting
$K_j = 1$) for all cells, and (2) one based directly on the likelihood of each map, without the state-space model. The results of approach
(1) are summarized in L7-13 of this response. By using approach (2), we observed that the decoded probabilities for each cell become
noisier, with means hovering around .5, and a large standard deviation of about 0.35. This low SNR results in increased uncertainty
about the represented environment, and could not be alleviated with simple approaches such as kernel methods. At the population level,
this noisiness drops significantly, but is still substantial, with an average standard deviation around 0.22. The average probability of
decoding the correct environment during the original, unambiguous environments for approaches (1) and (2) were computed and the
difference between these values and the same values when our state-space modeling approach is used is shown in Fig. 5C. This suggests
that the state space model improves decoding accuracy in 87% of cells. We observed these naive methods cannot yield similar results
shown in Fig. 4. A thorough comparison between these approaches and our state-space modeling approach will be added to our paper.

**R3: This paper has incremental technical contributions and the algorithm is a straightforward extension of previous state-**
**space models (HMMs).** While the model structure we used for the data analysis does fit into the general category of HMMs, the
algorithms for filtering, smoothing, covariance, and parameter estimation are not trivial and we believe that these derivations provide an
important contribution. The specification of this particular model structure also allowed us to uncover transitions in representations
that were previously undetected in this data. Finally, the general model structure we present (section 2.1) does not require the Markov
assumption (though we did assume this property in the specific analysis example here). For all these reasons, we believe that this work
provides an important contribution to the literature. We will add additional discussion of the generality of these methods to the revision.

**R3: The Gamma model in Eq. 6 is likely a poor model** Although in many cases simple Gamma models may perform poorly for
deconvolved Calcium response, our analyses suggest that this model fits very well to our data. This is demonstrated for two sample cells
in Fig. 5D. In addition, we performed a goodness-of-fit deviance analysis and found that the Gamma model was rejected for none of our
cells. Our revision will include a brief analysis of goodness-of-fit for this model to demonstrate this.

**R3, R4: The applicability of the proposed method and its potential applications in finding other qualitative theories.** While the
data example here focuses on estimating the cognitive representation of ambiguous environments, we believe that these methods are
broadly applicable to any situation where multiple representations of a stimulus or biological signal are present in a neural population.
Such multiple representations may reflect other factors, not controlled by the experiment, or may reflect alternate computational
principles within the brain. We believe that these methods provide statistical tools that will enable analyses of a wide range of
complex experiments involving multiple neural representations. For instance, one potential application is to detect sub-populations that
cooperatively represent an environment and study how these sub-populations change in response to alternation in the input stimuli.

Figure 5: Extra plots that summarize our new analyses.

[Meta-Review · NeurIPS 2020]

This paper proposes a model of how uncertainty and ambiguity are represented in neural activity, and validate on hippocampal CA1 recordings, collected on mice being exposed to different environments, It’s well-motivated and sure to be of interest to the NeurIPS neuroscience community. It appears to be well-written (R1, R2), contains excellent figures (R2, R3), and overall the review of prior relevant work is outstanding (R4), although R1 did bring up some missing prior work. The main novel contribution is the interesting theoretical framework of incorporating multiple representations for the same environment within a decoding model (R1, R4). However, R4 wondered about the wider theoretical implications for cognition in general, and whether it offers new insights beyond simply explaining the data. R1 raised some concerns about the correctness of the claims, given that the evidence for rapid fluctuations (one of the claims) is only apparent in single neurons, which can have low SNR, and not at the population level (although their other concerns were addressed and they raised score from 5 to 6). Despite several criticisms raised in review, the AC and SAC found the proposed methods to be highly innovative and likely to be of major interest to the computational neuroscience community focused on neural coding. They agreed that the paper makes a significant contribution, and recommend it for acceptance as a poster presentation.